# Visible Light-Based Ag_3_PO_4_/g-C_3_N_4_@MoS_2_ for Highly Efficient Degradation of 2-Amino-4-acetylaminoanisole (AMA) from Printing and Dyeing Wastewater

**DOI:** 10.3390/ijerph19052934

**Published:** 2022-03-02

**Authors:** Hong Liu, Houwang Chen, Ning Ding

**Affiliations:** 1Jiangsu Key Laboratory of Environmental Science and Technology, School of Environmental Science and Engineering, Suzhou University of Science and Technology, Suzhou 215009, China; houwangchen@cumt.edu.cn; 2Key Laboratory of Cleaner Production and Comprehensive Utilization of Resources, China National Light Industry, Department of Environmental Science and Engineering, Beijing Technology and Business University, Beijing 100000, China; dingning@btbu.edu.cn

**Keywords:** catalysis, degradation, nanomaterials, Ag_3_PO_4_, heterojunction formation

## Abstract

In this research, the preparation of a Ag_3_PO_4_/g-C_3_N_4_@MoS_2_ photocatalyst and the performance and mechanism of degradation of 2-amino-4-acetaminoanisole (AMA) were studied. The phase composition and morphology of the synthesized samples were comprehensively characterized by X-ray diffraction (XRD), scanning electron microscope (SEM), ultraviolet–visible diffuse reflectance (UV–Vis), and photoelectron spectroscopy (XPS). The catalytic performance of the photocatalyst was evaluated by the visible-light catalytic degradation of the AMA. The experimental results show that the Ag_3_PO_4_/g-C_3_N_4_@MoS_2_ composite photocatalyst has stronger photocatalytic oxidation and reduction capabilities than Ag_3_PO_4_ and Ag_3_PO_4_/g-C_3_N_4_. The material only decreases by 31.3% after five cycles of use, indicating that the material has good light stability. Free radical capture experiments prove that photo-generated holes (h^+^) and superoxide radicals (·O_2_^−^) are the main active substances in the photocatalytic process. The fundamental studies in the present research provide a new perspective for constructing an innovative type of visible-light photocatalyst and a new way to promote the photocatalytic degradation of organic pollutants.

## 1. Introduction

The treatment of industrial wastewaters is a historical challenge which is relevant and significant to the environment. Many industrial activities produce wastestream, which cannot be discharged directly into the environment [1,2]. Chemical industrial processes, in particular, generate wastewaters containing a wide variety of recalcitrant organic pollutants such as phenol, benzene, anilines, and chlorophenols, etc. Dyeing and printing processes are among the chemical processes that can also generate various types of wastewaters respective to the intermediate dyes or printing materials in use [3,4]. A dye intermediate with the IUPAC name 2-amino-4-acetaminoanisole (AMA) is widely used in textile, printing, and dyeing industries. The enormous use of this dye intermediate leads to the wastestream containing AMA. The AMA-dyed wastewater contains a large amount of waste and by-products’ waste liquid. The poor treatment of AMA-dyed wastestream could lead to serious water pollution [5]. At present, the common techniques for printing and dyeing wastewater treatment mainly include physical adsorption [6], coagulation [7], membrane separation [8], chemical oxidation [9], and biological methods [10]. However, the physical method faces the risk of causing secondary pollution, and the biological toxicity of dyes also restricts the development of the biological method in treating printing and dyeing wastewater.

Photocatalysis technology is an environmentally friendly technology for organic pollutants treatment. Among the most commonly applied photocatalytic materials, titanium dioxide has attracted wide attention because of its stability, non-toxicity, and low production cost. However, due to its low band gap (3.2 eV), which can only be excited by ultraviolet light, its catalytic applications are severely restricted [11,12,13,14,15,16]. Silver-based semiconductors, such as Ag_3_PO_4_ [17], AgI [18], AgBr [19], Ag_2_CO_3_ [20], and Ag_3_VO_4_ [21], etc., are photocatalysts with a visible-light response and can be used to degrade organic pollutants. Among them, the p-type semiconductor silver orthophosphate (Ag_3_PO_4_) has a high quantum yield of 90%, at a wavelength of 420nm, and is considered to be the best substitute for TiO_2_ [22]. However, the Ksp of silver phosphate is 1.6 * 10^−16^, and is easily corroded and decomposed into the silver element under light conditions (4Ag_3_PO_4_ + 6H_2_O + 12h^+^ + 12e^−^ → 12Ag^+^ 4H_3_PO_4_ + 3O_2_) [23,24]. Previous studies showed that these defects could be improved by a synthetic heterojunction structure, either through metal or non-metal doping and photosensitization [25]. Molybdenum disulfide (MoS_2_), as a catalyst, has tunable optical properties, a unique layered structure, and a large bandgap, and could be potentially applied in the degradation of organic pollutants [26,27,28,29,30,31,32], hydrogen release reactions (water decomposition), and CO_2_ reduction applications [33,34,35]. The edge potentials of the valence band and conduction band of bulk MoS_2_ and layered MoS_2_ are estimated to be 1.4 eV and 1.78 eV, respectively. Although MoS_2_ cannot directly produce free radicals to oxidize and decompose organic pollutants, it can be effectively used as a cocatalyst for the separation of photogenerated carriers. The control of the recombination rate of electron-hole carriers could be used to improve the photocatalytic activity of the composite material [36]. In addition, g-C_3_N_4_ is a highly stable and environmentally friendly material, which can be used to decompose organic pollutants under visible light [37]. It is an ideal photocatalyst that can be used to synthesize Ag_3_PO_4_/g-C_3_N_4_ photocatalytic materials.

Chang et al. reported lime coagulation combined with Fenton for AMA removal [4]. After 90 min, the removal rate of AMA reached 46%. However, there is no research on the photocatalytic decomposition of AMA, and the photocatalytic degradation mechanism of the difficult-to-degrade organic matter in AMA-dyed wastewater has not been clearly studied. The present paper attempts to prepare Ag_3_PO_4_/g-C_3_N_4_@MOS_2_ semiconductor composite materials. The AMA dye precursor was selected as the target contaminant for photodegradation. Therefore, the study objectives are (1) to test the photocatalytic activity of as-synthesized materials by degrading AMA; (2) to determine the stability and photo-corrosion resistance of materials through cycle experiments; and, finally, (3) to propose possible photocatalytic degradation pathways and mechanisms.

## 2. Experimental

### 2.1. Materials

All materials and chemicals were of analytical grade and were procured from different chemicals industries. These include silver nitrate (AgNO_3_, Shanghai Institute of Fine Chemical Materials, analytical pure, Shanghai, China); disodium hydrogen phosphate (Na_2_HPO_4_·12H_2_O, Runjie Chemical, guaranteed reagent, Shanghai, China); thiourea (CH_4_N_2_S, Runjie Chemical, analytical pure, Shanghai, China); melamine (C_3_H_6_N_6_, Sinopharm Chemical Reagent Co., Ltd., analytical pure, Beijing, China); molybdenum sodium (Na_2_MoO_4_, Tianjing Chemical Reagent No. 4 Factory, analytical pure, Tianjing, China); 2-amino-4-acetaminoanisole (AMA) (content >98%, Tokyo Chemical Industry Co., Ltd. Tokyo, Japan); absolute ethanol (Wuxi Jingke Chemical Co., Ltd., analytical pure, Wuxi, China); and ethylene glycol (Jiangsu Qiangsheng Functional Chemical Co., Ltd., analytical pure, Changshu, China). Acetonitrile, methanol, and dichloromethane were chromatographically pure and obtained from Wuxi Prospect Chemical Reagent Co., Ltd. In addition, scavengers such as p-benzene quinone (BQ), isopropanol (IPA) and ethylenediaminetetraacetic acid disodium salt (EDTA-2Na) were procured from Aladdin Reagent, Tianjin Bodi Chemical Co., Ltd., and Tianjin Guangfu Chemical Reagent Co., Ltd., respectively.

### 2.2. Synthesize of Ag_3_PO_4_/g-C_3_N_4_@MoS_2_

#### 2.2.1. Synthesis of g-C_3_N_4_ Flakes

Firstly, 20 g of melamine was weighed and poured into a ceramic crucible. The crucible was then transferred to a tubular atmosphere furnace and raised to 550 °C, at a heating rate of 5 °C·min^−1^, and kept for 3 h. After cooling to room temperature, the dried solid material was ground in a mortar and passed through a 100-mesh sieve to obtain a light yellow powder. The powder was spread in a porcelain boat. The porcelain boat was transferred to a tubular atmosphere furnace and raised to 500 °C, at a heating rate of 5 °C·min^−1^ for 2 h, and white g-C_3_N_4_ nanosheets were obtained after cooling.

#### 2.2.2. Synthesis of Ag_3_PO_4_

Stepwise, 50 mL of deionized water was measured and poured into a 100 mL beaker. Then, 0.764 g of silver nitrate solid was weighed and added to deionized water. The above mixture was stirred for 1 h until completely dissolved. Subsequently, a solution of disodium hydrogen phosphate was prepared by dissolving 0.537 g of disodium hydrogen phosphate in 50 mL of deionized water. The disodium hydrogen phosphate solution was gradually added dropwise to the silver nitrate solution and stirred for 2 h to fully react. After suction filtration, the yellow precipitate obtained from the reaction was transferred to a vacuum-drying oven at 60 °C for 12 h, ground into a powder to obtain the silver phosphate particles, and stored for later use.

#### 2.2.3. Synthesis of MOS_2_

To prepare MOS_2_, 7 mmol of sodium molybdate and 35 mmol of thiourea were weighed and dissolved in 160 mL of deionized water, stirred for 2 h until fully dissolved, then transferred to a Teflon autoclave. The reactor was placed in an oven to set the temperature to 200 °C for 24 h. After cooling, the material obtained was washed with deionized water, collected by centrifugation, and dried in a vacuum-drying oven at 60 °C for 12 h. The final product was stored for later use.

#### 2.2.4. Synthesis of Ag_3_PO_4_/g-C_3_N_4_@MoS_2_

Absolute ethanol (40 mL) and ethylene glycol (2 mL) were poured into a beaker and then 0.2 g of g-C_3_N_4_ was added to the solution. The above solution was ultrasonicated for 2 h and recorded as solution A. Subsequently, 8 mg of MoS_2_ was added to 40 mL of deionized water, sonicated for 2 h, and recorded as solution B. Additionally, another solution, recorded as C, was prepared by adding 0.764 g of silver nitrate solid to deionized water and stirred for 1 h to dissolve it completely. Moreover, 0.537 g of disodium hydrogen phosphate was dissolved in 50 mL of deionized water, stirred for 1 h, and recorded as solution D. Solution A and solution B were added dropwise to solution C and stirred for 2 h. Finally, solution D was added dropwise and stirred for 2 h to fully react. After suction filtration, the precipitate obtained from the reaction was dried in a vacuum-drying oven at 60 °C for 12 h, and ground to a powder to obtain the Ag_3_PO_4_/g-C_3_N_4_@MoS_2_ composite material.

### 2.3. Characterization of Ag_3_PO_4_/g-C_3_N_4_@MoS_2_

The phase analysis of the material was carried out by an X’Pert PRO (Panalytical, The Netherlands) X-ray diffractometer, using Cu target radiation, a test voltage of 40 kV, a current of 40 mA, and a scanning range of 10°~90°. The microscopic morphology of the samples was observed using an SU8010/S4800 (Japan/Hitachi) high-resolution cold field emission scanning electron microscope (SEM) with a working voltage of 1 kV and a resolution of 1.3 nm. The surface element composition and element valence analysis of the samples were tested with an (ESCALAB 250xi) (U.S. Thermo, Waltham, MA, USA) X-ray photoelectron spectrometer. The ultraviolet–visible diffuse reflectance map was generated by a Shimadzu UV3600 ultraviolet (Kyoto, Japan)–visible spectrophotometer with a scanning range of 200–800 nm. A VERTEX70 (Ettlingen, Germany) infrared spectrometer, produced by Bruker, was used to analyze the infrared spectrum of the material. The measurement range was 30,000–10 cm^−1^, the resolution was 0.4 cm^−1^, and the measurement accuracy was 0.1. The photoluminescence spectra were detected by a fluorescence spectrophotometer (Edinburgh steady/transient fluorescence spectrometer). The conventional platen method was used for the excitation wavelength of 350 nm.

### 2.4. Photocatalytic Activity Tests

The photocatalytic activity of Ag_3_PO_4_/g-C_3_N_4_@MoS_2_ was tested by the photocatalytic degradation of AMA at 25 °C. The influence of MoS_2_ on the photocatalytic degradation activity was studied by adding various MoS_2_ contents. In addition, to evaluate the photocatalytic activity of Ag_3_PO_4_/g-C_3_N_4_@MoS_2_, the photocatalytic activity of Ag_3_PO_4_, Ag_3_PO_4_/g-C_3_N_4,_ and Ag_3_PO_4_/g-C_3_N_4_@MoS_2_ were used as comparison. A slurry reactor was used by adding 0.03 g of the photocatalyst to 20 mg/L of the AMA solution. The solution was, first, magnetically stirred in a dark environment for 30 min to obtain a good dispersion and adsorption equilibrium between the substance and the catalyst surface. After the solution was irradiated with a 300 w xenon lamp. During the photodegradation course, a 1 mL sample was drawn from the solution every 10 min, filtered with a 0.22 μm filter and a syringe. The concentration of AMA was monitored by HPLC. The rate of AMA degradation by the catalyst was estimated according to Formula (1).
(1)$$\mathrm{Degradation}\;\mathrm{rate}\%=(1-\frac{C_{t}}{C_{0}})\;\times \;100\%$$
where *C*_0_ and *C_t_* are the initial concentration of the target AMA and the concentration of AMA at time t after the start, respectively.

The chromatographic conditions of HPLC were as follows: a Zorbax Eclipse Plus C18 chromatographic column (column length 150 mm, inner diameter 4.6 mm, particle size 3.5 μm), equipped with a UV detector with a detection wavelength of 254 nm and a column temperature of 40 °C. For the mobile phase, the methanol/ammonium acetate buffer was 33/67, the ammonium acetate buffer was 1 g/L, with a flow rate of 1.2 mL/min, and an injection volume of 5 μL.

### 2.5. Anti-Photo Corrosion Evaluation

To test the corrosion resistance of Ag_3_PO_4_/g-C_3_N_4_@MoS_2_, cycling experiments were performed. In a typical procedure, 0.03 g of the composite material was added to 20 mg/L (50 mL) of AMA solution, and samples were taken every 10 min. High-performance liquid chromatography (HPLC) was employed to detect the concentration of AMA. After 1 h of illumination, the materials in the reactor were collected by centrifugation, filtered, washed three times with absolute ethanol and deionized water, and transferred to a vacuum-drying oven for 12 h at 60 °C. The dried materials were then stored for later use.

### 2.6. Photocatalytic Degradation Mechanism of AMA

Intermediate products in the process of the photocatalytic degradation of AMA were detected by high-performance liquid chromatography–mass spectrometry (HPLC) technology. Free radical capture experiments were carried out to detect the main active substances and oxidized substances in the reaction process. Isopropanol (IPA) (1 mM), p-benzoquinone (BQ) (1 mM), and disodium ethylenediaminetetraacetic acid (EDTA-2Na) (1 mM) were used to detect the hydroxyl radicals (·OH), superoxide radicals (·O_2_^−^), and holes (h^+^), respectively. Stepwise, 0.03 g of the composite material was added to 50 mL of the 20 mg/L AMA solution. To the above solution, 1 mmol of either IPA, BQ, and/or EDTA-2Na was added. Samples were instantaneously taken every 10 min, and the concentration of AMA in the samples was detected by high-performance liquid chromatography.

## 3. Results and Discussion

### 3.1. Characterization of Ag_3_PO_4_/g-C_3_N_4_@MoS_2_

#### 3.1.1. SEM Analysis

The morphologies of Ag_3_PO_4_, g-C_3_N_4_, MoS_2_, and Ag_3_PO_4_/g-C_3_N_4_@MoS_2_ were determined by SEM images (Figure 1). From Figure 1a, it can be seen that Ag_3_PO_4_ is an irregular particle with a diameter of about 500 nm. The as-prepared g-C_3_N_4_ in Figure 1b has a lamellar structure, which facilitates the transfer of electrons. The MoS_2_ in Figure 1c shows a thin nanosheet structure. The SEM image of the composite Ag_3_PO_4_/g-C_3_N_4_@MoS_2_ is shown in Figure 1d. The Ag_3_PO_4_ is evenly distributed on the surface of the MoS_2_ and g-C_3_N_4_, and the size of Ag_3_PO_4_ is smaller, approximately 200–400 nm.

#### 3.1.2. X-ray Diffraction (XRD)

X-ray detection of Ag_3_PO_4_, g-C_3_N_4_, MoS_2,_ and the crystal structure of composite materials are shown in Figure 2a. The synthesized Ag_3_PO_4_ shows a cubic crystal structure as compared with the standard card JCPDS No. 06-0505, and there is no impurity peak, indicating that the synthesized Ag_3_PO_4_ has a high purity. Further, two obvious diffraction peaks of the g-C_3_N_4_ flakes can be observed at 2θ of 12.6° and 27.5°, corresponding to the (100) and (002) crystal planes of the g-C_3_N_4_, respectively. While, for the composite material Ag_3_PO_4_/g-C3N4@MoS_2_ shown in Figure 2b, the diffraction peaks corresponding to the cubic Ag_3_PO_4_ and the characteristic peaks of the g-C_3_N_4_ can be observed, the characteristic peaks of the MoS_2_ are not observed, which may be attributed to the low loading of MoS_2_.

#### 3.1.3. X-ray Photoelectron Spectroscopy (XPS)

The chemical composition and chemical state of the surface of the composite Ag_3_PO_4_/g-C_3_N_4_@MoS_2_ were analyzed by XPS spectra detection. It can be seen from Figure 3a that the composite material has Ag, P, O, C, N, Mo, and S elements. Figure 3b shows the two peaks of Ag 3d at 368.1 eV and 374.1 eV, which correspond to the Ag 3d_5/2_ orbital and Ag 3d_3/2_ orbital of Ag^+^. The peak in the XPS spectrum of P 2p, shown in Figure 3c, is located at 132.3 eV, which belongs to the P^5+^ in silver phosphate. The high-resolution characteristic peaks of O 1s, as shown in Figure 3d, are fitted to two characteristic peaks of 530.7 eV and 532.8 eV, which can be attributed to the O^−^ and hydroxyl groups in silver phosphate. There are three characteristic peaks of XPS of N 1s in the composite material at 399 eV, 400.3 eV, and 401.3 eV (Figure 3e). These peaks can be attributed to the sp^2^ hybrid carbon in C=N-C, bridging the nitrogen N-(C)^3^ and the amino functional group C-N-H (Figure 3f). Three characteristic peaks can be seen in the XPS graph of Mo 3d shown in Figure 3g. The peaks at 229.9 eV and 232.8 eV correspond to Mo 3d_5/2_ and Mo 3d_3/2_, respectively, which indicates the presence of Mo^4+^ in the synthesized molybdenum disulfide. The characteristic peak at 235.8 eV is attributed to Mo^6+^. There are two characteristic peaks of 162.7 eV and 164 eV in the S 2p spectrum, as shown in Figure 3h, which can correspond to the S 2p_1/2_ and S 2p_3/2_ in MoS_2_. The XPS chart proves that there are three materials: silver phosphate, graphite carbon nitride, and molybdenum disulfide in the composite material, which are consistent with the XRD results.

#### 3.1.4. Fourier Infrared Spectroscopy Analysis (FT-IR)

Figure 4 shows the Fourier infrared spectra of Ag_3_PO_4_, g-C_3_N_4_, MoS_2_, Ag_3_PO_4_/g-C_3_N_4_, and Ag_3_PO_4_/g-C_3_N_4_@MoS_2_. Among them, the spectrum of silver phosphate has two characteristic peaks at 559 cm^−1^ and 1012 cm^−1^, which is due to the stretching vibration of the P-O bond in PO_4_^3-^. The characteristic peak of g-C_3_N_4_ at 811 cm^−1^ is produced by the vibration of the tri-S-triazine ring in g-C_3_N_4_. The characteristic peaks at 1240 cm^−1^, 1309 cm^−1^, and 1575 cm^−1^ may be caused by the stretching vibration of the C=N or C-N bond.

#### 3.1.5. UV/vis Diffuse Reflectance Spectrum

To further investigate the photocatalytic properties of Ag_3_PO_4_/g-C_3_N_4_@MoS_2_, Ag_3_PO_4_, g-C_3_N_4,_ and MoS_2_, UV/vis diffuse reflectance spectroscopy was employed. As shown in Figure 5a, compared with single silver phosphate and g-C_3_N_4_, the composite Ag_3_PO_4_/g-C_3_N_4_@MoS_2_ has a stronger visible-light absorption ability in the 200–800 nm wavelength range, and the absorbance in the visible-light region is also enhanced (Figure 5b). Using Formula (2), we determined the bandgap widths of Ag_3_PO_4_, g-C_3_N_4,_ and MoS_2_, estimated to be 2.13 eV, 3.08 eV, and 1.78 eV, respectively (Figure 5c).
(2)$$\mathrm{Avh}\;=\;A(\mathrm{hv}\;-\;\mathrm{Eg}{)}^{n/2}$$

A is a constant, hv is the photon energy, h is the Plane g constant, v is the frequency, Eg represents the semiconductor band gap and n is an index. Here, n is related to the type of semiconductor; n of the direct bandgap semiconductor is 1/2, and n of the indirect bandgap semiconductor is 2 [38].

According to the following equations of ECB = χ − Ee − 0.5 Eg and EVB = ECB + Eg, the conduction bands of Ag_3_PO_4_, g-C_3_N_4,_ and MoS_2_ were calculated to be +0.72 eV, −1.68 eV, and 0.02 eV, respectively. The valence bands of Ag_3_PO_4_, g-C_3_N_4,_ and MoS_2_ were calculated to be +2.85 eV, +1.4 eV, and +1.8 eV, respectively (Figure 5d).

### 3.2. Photocatalytic Degradation of AMA

The photocatalytic performance of Ag_3_PO_4_, Ag_3_PO_4_/g-C_3_N_4_, Ag_3_PO_4_/g-C_3_N_4_@MoS_2,_ and different composite ratios was evaluated by the photocatalytic degradation of AMA. As shown in Figure 6a, after 30 min of dark adsorption, the adsorption capacity of the Ag_3_PO_4_/g-C_3_N_4_ composite and Ag_3_PO_4_/g-C_3_N_4_@MoS_2_ were both stronger than that of the single silver phosphate, indicating that the combination of materials is beneficial to the adsorption of pollutants. In the photocatalytic degradation stage, it can be seen from Figure 6a that, with the increase in the C_3_N_4_ loading, the photocatalytic effect of Ag_3_PO_4_/g-C_3_N_4_ gradually increased. When the loading amount reached 0.2 g, the photocatalytic activity of Ag_3_PO_4_/g-C_3_N_4_ was at its best, with 95.9% AMA degradation occurring in 60 min, after which, with the increase in the g-C_3_N_4_ dosage, the photocatalytic effect began to decrease. When the loading amount reached 0.4 g, the composite tube catalytic effect was less than the photocatalytic effect of the silver phosphate monomer. Combining an appropriate amount of g-C_3_N_4_ with Ag_3_PO_4_ can significantly improve the photocatalytic ability, and the amount of g-C_3_N_4_ plays a crucial role in the composite material. When the amount of g-C_3_N_4_ is too small, Ag_3_PO_4_ cannot be evenly distributed on the g-C_3_N_4_, and the ability of g-C_3_N_4_ to transfer electrons is weak; the ability of Ag_3_PO_4_ to capture photons is weakened and too much g-C_3_N_4_ may form recombination centers for photogenerated electron holes.

As depicted in Figure 6b, when the dose of MoS_2_ was increased to 8 mg, the photocatalytic performance was at its best. Then, as the dosage increases gradually, the photocatalytic activity is gradually weakened. Few-layer MoS_2_ played a small role in the separation of electron–hole pairs, causing the loss of oxidation ability due to the decline in Ag_3_PO_4_ content. If the amount of MoS_2_ is too large, MoS_2_ weakens the ability of Ag_3_PO_4_ to capture photons due to its shielding effect.

### 3.3. Anti-Photo Corrosion Evaluation

In the practical application of the photocatalytic degradation of pollutants, whether the catalyst can be recycled is one of the important indicators to judge the performance of photocatalytic materials. In this experiment, the stability of Ag_3_PO_4_/g-C_3_N_4_@MoS_2_ was judged by multiple cycles. The experimental results are shown in Figure 7. After five cycles of recycling, the degradation efficiency was only reduced by 31.3%, indicating that the Ag_3_PO_4_/g-C_3_N_4_@MoS_2_ material has good stability and can be recycled at least five times.

### 3.4. Photocatalytic Degradation Pathway of AMA

#### 3.4.1. Free Radical Capture Experiment

In the photocatalytic degradation of AMA using Ag_3_PO_4_/g-C_3_N_4_@MoS_2_, isopropanol (IPA), p-benzoquinone (BQ), and disodium ethylenediaminetetraacetic acid (EDTA-2Na) were added to capture h^+^, ·OH, and ·O_2_^−^, respectively. The experimental results are depicted in Figure 8.

Among the three capture agents added, the addition of isopropanol had almost no effect on the degradation of AMA, while the addition of p-benzoquinone and ammonium oxalate inhibited the photocatalytic efficiency to a greater extent, reaching 36% and 70%, respectively. The results show that superoxide radicals and holes play a major role in the process of catalyzing and oxidizing organics.

#### 3.4.2. Proposed Photocatalytic Degradation Mechanism

Based on the aforementioned results of the free radical capture experiments and the positions of the valence and conduction bands of Ag_3_PO_4_, g-C_3_N_4,_ and MoS_2_ obtained by calculation, we analyzed the possible photocatalytic degradation mechanism of AMA by Ag_3_PO_4_/g-C_3_N_4_@MoS_2_ (Figure 9). Under visible-light irradiation, Ag_3_PO_4_ and g-C_3_N_4_ are excited, and generate photo-generated electrons and holes in the conduction band and valence band, respectively. Although the large band gap of g-C_3_N_4_ (3.08 eV) will limit the rate of the whole photocatalysis, its suitable band-gap width is just enough to form a Z-type heterojunction composite with Ag_3_PO_4_ and MoS_2_. Meanwhile, its wide band gap is beneficial to electron transfer, with the reduction in the recombination of electrons and holes, which can enhance the photocatalytic effect. As the conduction-band potential of Ag_3_PO_4_ of 0.65 eV is not enough to reduce O_2_ to ·O_2_^−^(−0.33 eV), ·O_2_^−^ is mainly produced by the conduction band of MoS_2_. Photogenerated electrons are generated by the conduction band of Ag_3_PO_4_, transferred to the valence band of MoS_2_, and then transferred to the conduction band of MoS_2_ by light excitation. The resulting photogenerated electrons can reduce O_2_ to produce ·O_2_^−^, which can be used to decompose AMA. The holes of Ag_3_PO_4_ can be directly used to oxidize AMA, which can further be decomposed into the simple and non-toxic molecules of H_2_O and CO_2_.

#### 3.4.3. Analysis of Photocatalytic Degradation Products of AMA

Under visible-light irradiation, a variety of active substances (including h^+^, ·OH, and ·O_2_^−^) on the catalyst surface can interact with AMA. During the reaction, the molecular structure of AMA is destroyed and some intermediate products are produced. The possible degradation pathways were proposed based on HPLC-MS analysis shown in Figure 10 and Figure 11. As shown in Figure 10a, before the start of the reaction, only AMA is present in the solution with a mass-to-charge ratio of 181. After 20 min of light irradiation, as shown in Figure 10b, the amino group in the AMA is first oxidized by holes (h^+^) to form a compound with an m/*z* of 198. With the increase in the reaction time after 40 min of irradiation, as shown in Figure 10c, a nitro group on the benzene ring is oxidized and falls off to form a compound with an m/*z* of 150. With the further chemical reaction, the methoxy functional group is oxidized to hydroxyl, and the compound is converted from ethers to phenols. After 60 min of light illumination (Figure 10d), the hydroxyl and nitro groups in the compound is gradually oxidized and falls off, and the benzene ring is also opened during the oxidation process. Finally, with continuous visible-light irradiation, the above-mentioned intermediate products are mineralized into H_2_O and CO_2_.

## 4. Conclusions

The Ag_3_PO_4_/g-C_3_N_4_@MoS_2_ composites with different MoS_2_ contents were successfully synthesized by electrostatic self-assembly and ion-exchange methods. The as-synthesized Ag_3_PO_4_/g-C_3_N_4_@MoS_2_ showed a more efficient visible-light degradation performance. Composite materials can effectively reduce the recombination rate of photo-generated carriers, thereby improving photocatalytic activity. Free radical capture experiments were performed by using isopropyl alcohol (IPA), p-benzoquinone (BQ), and disodium ethylenediaminetetraacetic acid (EDTA-2Na) like scavengers. It was observed that holes and superoxide radicals are the main active substances in the photocatalytic degradation reaction. During the photocatalytic degradation of Ag_3_PO_4_/g-C_3_N_4_@MoS_2,_ AMA was effectively degraded into simple molecules (g. CO_2_ and H_2_O) and some intermediate products, such as anisole and resorcinol. The analysis of the intermediate products further confirms that holes are the main active substances in the reaction. This research proposed a method to improve the catalytic performance of Ag_3_PO_4_, and also provides a way to improve the treatment of refractory organic pollutants by photocatalytic reagent.

## Figures and Tables

**Figure 1 ijerph-19-02934-f001:**
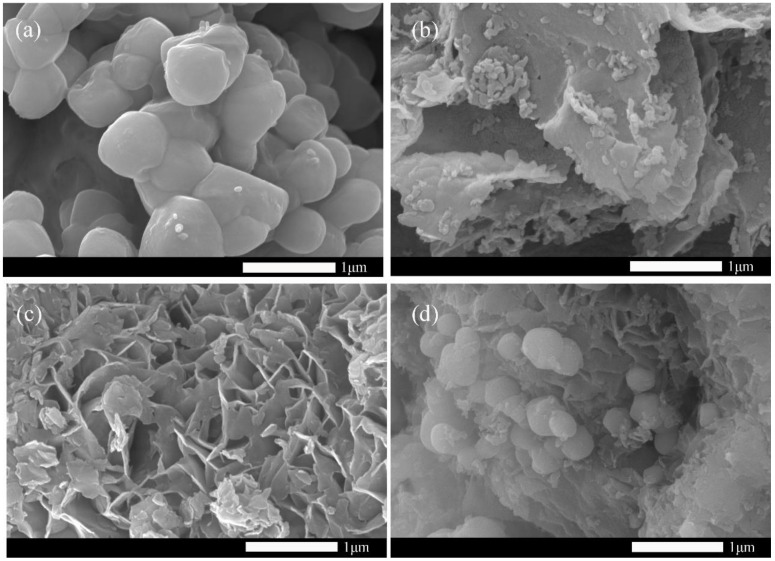
SEM pictures of Ag_3_PO_4_ (**a**), g-C_3_N_4_ (**b**), MoS_2_ (**c**), and Ag_3_PO_4_/g-C_3_N_4_@MoS_2_ (**d**).

**Figure 2 ijerph-19-02934-f002:**
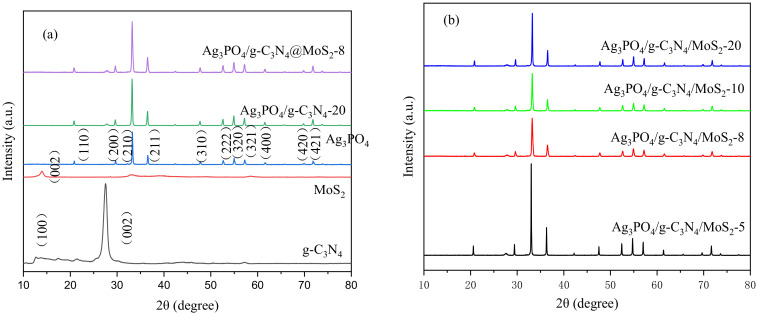
XRD patterns of Ag_3_PO_4_, g-C_3_N_4_, and MoS_2_ (**a**), and the Ag_3_PO_4_/g-C_3_N_4_@MoS_2_ composites (**b**).

**Figure 3 ijerph-19-02934-f003:**
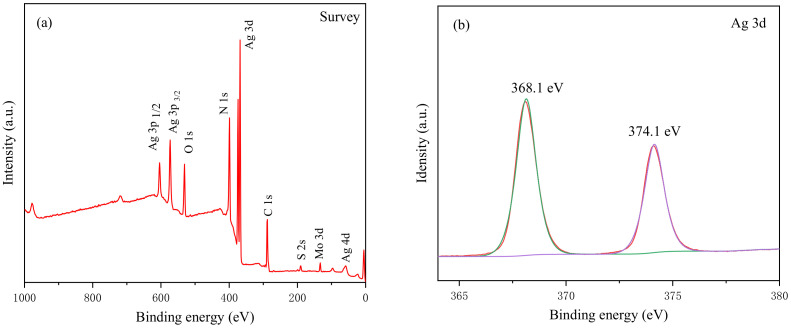
XPS images of survey spectra (**a**), Ag (**b**), P (**c**), O (**d**), N (**e**), C (**f**), Mo (**g**), and S (**h**).

**Figure 4 ijerph-19-02934-f004:**
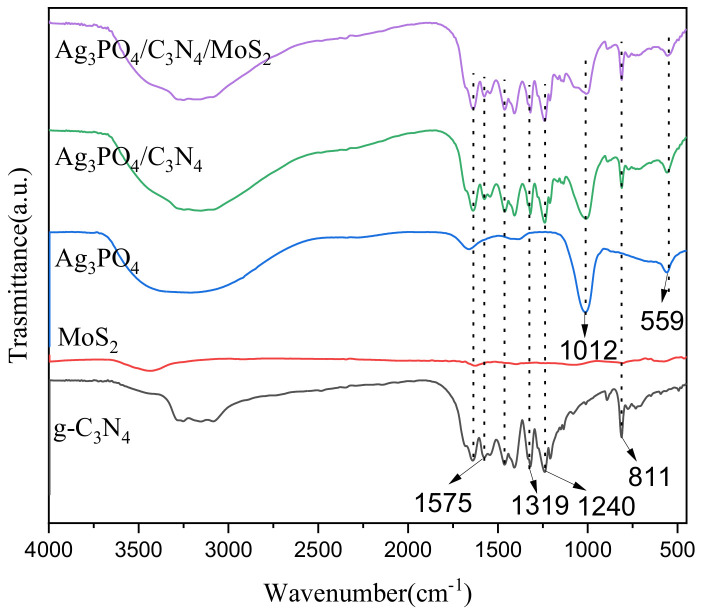
FT-IR spectra of Ag_3_PO_4_, g-C_3_N_4_, MoS_2_, Ag_3_PO_4_/g-C_3_N_4,_ and Ag_3_PO_4_/g-C_3_N_4_@MoS_2_.

**Figure 5 ijerph-19-02934-f005:**
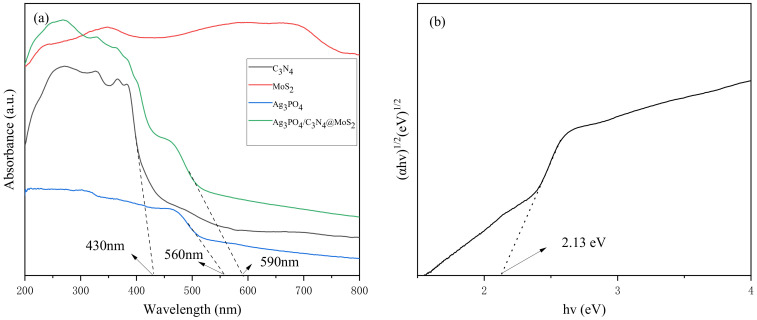
(**a**) UV–vis DRS; (**b**) curves of (αhv)^1/2^ versus hv of Ag_3_PO_4_; (**c**) (αhv)^2^ versus hv of g-C_3_N_4_ and MoS_2_; and (**d**) valence band XPS picture.

**Figure 6 ijerph-19-02934-f006:**
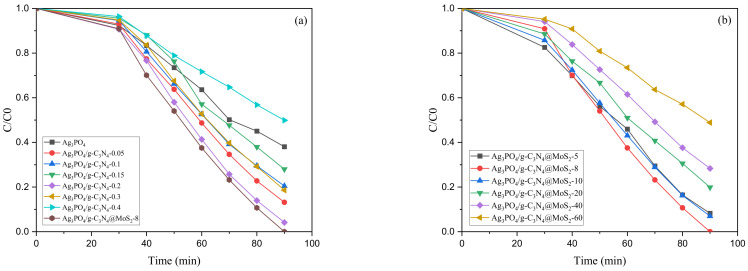
(**a**) degradation efficiency with different loadings of C_3_N_4_; (**b**) degradation efficiency with different dose of MoS_2_.

**Figure 7 ijerph-19-02934-f007:**
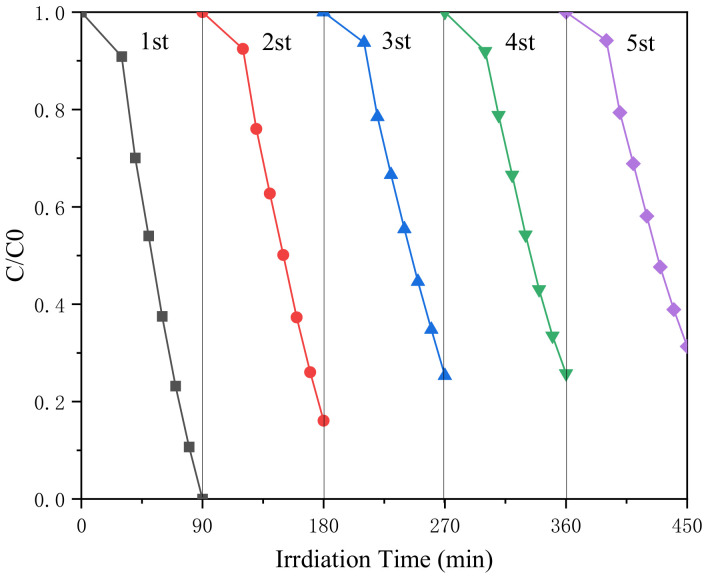
Regeneration cycles of the Ag_3_PO_4_/g-C_3_N_4_@MoS_2_-8 heterostructure.

**Figure 8 ijerph-19-02934-f008:**
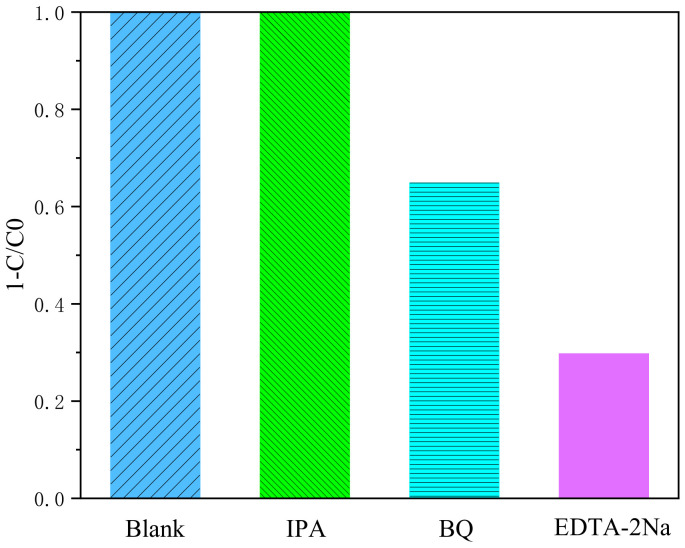
Degradation rate of AMA with different scavengers.

**Figure 9 ijerph-19-02934-f009:**
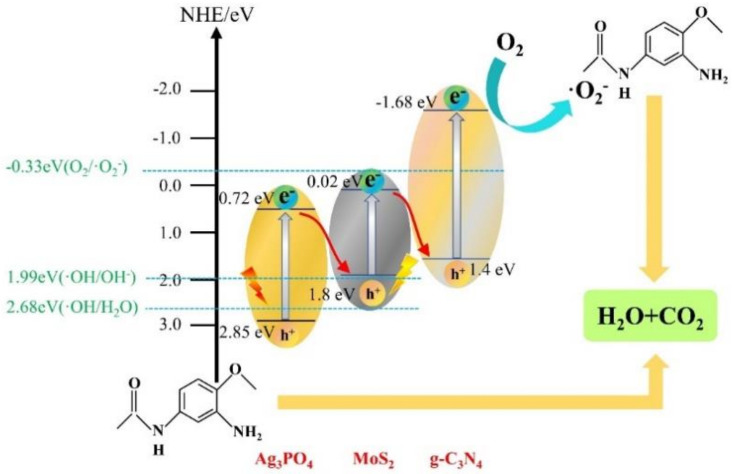
Proposed charge mechanism.

**Figure 10 ijerph-19-02934-f010:**
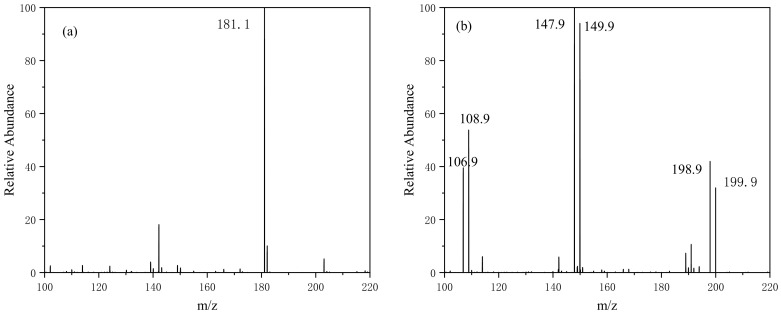
The mass spectrum of Ag_3_PO_4_/g-C_3_N_4_@MoS_2_ photocatalytic degradation of AMA: 0 min (**a**); 20 min (**b**); 40 min (**c**); and 60 min (**d**).

**Figure 11 ijerph-19-02934-f011:**
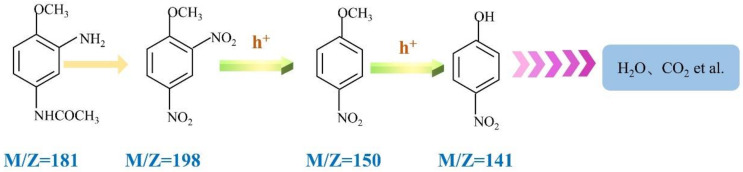
Possible degradation pathways of Ag_3_PO_4_/g-C_3_N_4_@MoS_2_ photocatalytic degradation of AMA.

## Data Availability

Not applicable.

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
