# Peer review of "Visible Light-Based Ag3PO4/g-C3N4@MoS2 for Highly Efficient Degradation of 2-Amino-4-acetylaminoanisole (AMA) from Printing and Dyeing Wastewater"

_ijerph, 2022, doi:10.3390/ijerph19052934_

Round 1
Reviewer 1 Report
The manuscript “ijerph-1579565” entitled “Visible light based Ag3PO4/g-C3N4@MoS2 for highly efficient degradation of 2-amino-4-acetylaminoanisole (AMA) from printing and dyeing wastewater” deals with the photocatalytic degradation of organic pollutant i.e. AMA from wastewater. As a general comment, the researchers have put in their efforts in providing the description of their results. However, there are many areas and results where scientific reasoning, unanswered why’s justification and discussion is lacking, which needs serious attention. Some of my comments are as follows:
- Why researchers used AgPO4 why not Ag2CO3 or Ag3VO4? What if we compare other silver-based catalysts for degradation organics? Further why not iridium-based catalyst? State the reason why you prefer or focus on silver?
- The researchers have rarely discussed about degradation of AMA. Is that studies on its photocatalytic degradation doesn’t exist and is that it’s not that much problematic pollutant? Further adding what is its guideline value? Is there any lower limit for its permissible discharge as per regulatory authority standards? All these aspects are unanswered in current manuscript.
- Line 159! Why researchers noted degradation potential of their catalysts for time duration of total of 90 minutes. If I look at various degradation results graphs, it seems if we provide more time photocatalytic degradation of AMA can be achieved up to 100% in all synthesized catalyst material. What is researchers exact motive in this regard.
- Besides SEM images gives good indication of synthesized catalyst material, there is still EDX data missing in your results. EDX is always good for providing the complete picture regarding the % weight and % atomic composition of individual element.
- Line 210! The researchers claim that while synthesizing composite catalyst material, XRD result didn’t indicate characteristics MoS2 peak in their composite due to its low concentration used for composite preparation purpose. Here comes a dire need to must to present EDX data as well as to do correction in their XRD results by either reanalysis.
- Figure 2B! Again how it is justified that researchers continue to increase MoS2 concentration and still no characteristics peak of MoS2 is observed. Needs serious attention.
- Section 3.1.5.! This section requires elaboration of each individual parameter. If researchers are using Tauc equation, they must refer it and add its complete details in supplementary information.
- “Line 274-277: When the loading amount reaches 0.2g, the photocatalytic activity of Ag3PO4/g-C3N4 is the best, with 95.9% AMA degradation in 60 minutes, after which, with the increase of g-C3N4 dosage, the photocatalytic effect began to decrease. When the loading amount reached 0.4g.” Why is it so? Researchers need to explain?
- Line 279-281 “when the dose of MoS2 was increased to 8mg, the photocatalytic performance showed the best, and then as the dosage increases gradually, the photocatalytic activity is gradually weakened”. Why is it so? Explain!!
- Section 3.4.1. Complete reasoning is required. These are key points that require scientific and justified reasoning. The researchers just reported and then moves to other section. This point needs improvement.
- Line 336-337! How can researchers say that the above-mentioned intermediate products are mineralized into H2O and CO2. Is there analysis done by researchers or are they referring to other study or is there any solid proof. Complete information is required in this regard?
Else, as a minor comment, references are not set as per journal format.
Author Response
Responses to the Editor and Reviewers’ Comments
Manuscript ID: IJERPH-1579565
Title: “Visible light based Ag3PO4/g-C3N4@MoS2 for highly efficient degradation
of 2-amino-4-acetylaminoanisole (AMA) from printing and dyeing wastewater”
Author(s): Hong Liu*, Houwang Chen, Ning Ding
We appreciate the time and effort of the Reviewers and the Editor in reviewing our manuscript. The reviews are very helpful for us to improve the manuscript. As a result of the comments from both the Editor and the Reviewers we have made significant changes and have rewritten parts of the manuscript. Point to point respond to all comments are as follows. Revised parts are marked in RED color in the revised manuscript.
Reviewer #1:
Comments:
The manuscript “ijerph-1579565” entitled “Visible light based Ag3PO4/g-C3N4@MoS2 for highly efficient degradation of 2-amino-4-acetylaminoanisole (AMA) from printing and dyeing wastewater” deals with the photocatalytic degradation of organic pollutant i.e. AMA from wastewater. As a general comment, the researchers have put in their efforts in providing the description of their results. However, there are many areas and results where scientific reasoning, unanswered why’s justification and discussion is lacking, which needs serious attention. Some of my comments are as follows:
Why researchers used AgPO4 why not Ag2CO3 or Ag3VO4? What if we compare other silver-based catalysts for degradation organics? Further why not iridium-based catalyst? State the reason why you prefer or focus on silver?
Response:
We thank for the reviewer’s suggestions. Ag3PO4, Ag2CO3, and Ag2CO3 all have good photocatalytic activity. In previous studies, the researchers found: Among them, Ag3PO4 are considered to have the highest quantum efficiency of 90% at the wavelengths longer than 420 nm(Tang C, Liu E, Wan J, et al. Co3O4 nanoparticles decorated Ag3PO4tetrapods as an efficient visible-light-driven heterojunction photocatalyst[J]. Applied Catalysis B: Environmental, 2016, 181: 707-715.)Therefore, we mainly study Ag3PO4 in this experiment. Precious metals such as iridium, gold, and platinum are also used in the field of catalysis, but considering the cost, we prefer silver-based catalysts. In the follow-up research, we will also consider using iridium-based catalysts for photocatalytic experiments.
Comments:
The researchers have rarely discussed about degradation of AMA. Is that studies on its photocatalytic degradation doesn’t exist and is that it’s not that much problematic pollutant? Further adding what is its guideline value? Is there any lower limit for its permissible discharge as per regulatory authority standards? All these aspects are unanswered in current manuscript.
Response:
We appreciate for the reviewer’s comments. AMA is a dye commonly used in dyes and other industries, and the hazard of AMA belonging to aniline is unknown. Although there are few existing studies, it cannot be ignored. The emission standard of aniline is 1mg/L. and the initiation of our research was that a local industrial company came to find us to solve this problem that they are facing, and we believe that our research provides important theoretical basis to similar wastewater industry.
Comments:
Line 159! Why researchers noted degradation potential of their catalysts for time duration of total of 90 minutes. If I look at various degradation results graphs, it seems if we provide more time photocatalytic degradation of AMA can be achieved up to 100% in all synthesized catalyst material. What is researchers exact motive in this regard.
Response:
We thank the reviewer for these precious comments and suggestions. When the experiment reaches 90 minutes, the degradation rates of the experimental materials can already be shown, and Ag3PO4/g-C3N4@MoS2 reached the reaction end point at 90 minutes already. So we set the time to 90 minutes. We agree that if the time is long enough, all catalytic materials can achieve 100% photocatalytic degradation of AMA, but the main purpose of this experiment is to screen out the materials with the fastest degradation rate.
Comments:
Besides SEM images gives good indication of synthesized catalyst material, there is still EDX data missing in your results. EDX is always good for providing the complete picture regarding the % weight and % atomic composition of individual element. And Line 210! The researchers claim that while synthesizing composite catalyst material, XRD result didn’t indicate characteristics MoS2 peak in their composite due to its low concentration used for composite preparation purpose. Here comes a dire need to must to present EDX data as well as to do correction in their XRD results by either reanalysis. Figure 2B! Again how it is justified that researchers continue to increase MoS2 concentration and still no characteristics peak of MoS2 is observed. Needs serious attention.
Response:
We detected the EDX data during the test as shown in the following figures, the Ag3PO4/g-C3N4@MoS2 composite can be seen in the SEM image, but the content of MoS2 is low in EDX, so it is difficult to reflect the specific percentage content in the data.
Comments:
Section 3.1.5.! This section requires elaboration of each individual parameter. If researchers are using Tauc equation, they must refer it and add its complete details in supplementary information.
Response:
We appreciate for the reviewer’s comments. We have supplemented each individual parameter in Section 3.1.5
Comments:
“Line 274-277: When the loading amount reaches 0.2g, the photocatalytic activity of Ag3PO4/g-C3N4 is the best, with 95.9% AMA degradation in 60 minutes, after which, with the increase of g-C3N4 dosage, the photocatalytic effect began to decrease. When the loading amount reached 0.4g.” Why is it so? Researchers need to explain?
Response:
We appreciate for the reviewer’s comments. Combining an appropriate amount of g-C3N4 with Ag3PO4 can significantly improve the photocatalytic ability, and the amount of g-C3N4 plays a crucial role in the composite material, when the amount of g-C3N4 is too small, Ag3PO4 cannot be evenly distributed on g-C3N4, and the ability of g-C3N4 to transfer electrons is weak; The ability of Ag3PO4 to capture photons is weakened, and too much g-C3N4 may form recombination centers for photogenerated electron holes. We have added these discussions in the revised manuscript. Please refer to page 14.
Comments:
Line 279-281 “when the dose of MoS2 was increased to 8mg, the photocatalytic performance showed the best, and then as the dosage increases gradually, the photocatalytic activity is gradually weakened”. Why is it so? Explain!!
Response:
We appreciate for the reviewer’s comments. Few-layer MoS2 played a small role in separation of electron-hole pairs, causing the loss of oxidation ability due to the decline of Ag3PO4 content. If the amount of MOS2 is too large, MoS2weakens the ability of Ag3PO4 to capture photons due to the shielding effect.
We have added the discussion in Page 15.
Comments:
Line 336-337! How can researchers say that the above-mentioned intermediate products are mineralized into H2O and CO2. Is there analysis done by researchers or are they referring to other study or is there any solid proof. Complete information is required in this regard?
Response:
We appreciate for the reviewer’s comments.
This conclusion was drawn based on previous research. The products of the photocatalytic degradation on organic matter has been detected. Results showed that photocatalysis can effectively mineralize the intermediate products into H2O and CO2. (Zhang, Z.Z.; Pan, ZW.; Guo, Y.F Wong. P.K.; Zhou, X.J.; Bai, R.B. In-situ growth of all-solid Z-scheme heterojunction photocatalyst of Bi7O9I3 /g-C3N4 and high efficient degradation of antibiotic under visible light. Applied Catalysis B: Environmental, 2020, 261).
We have added the above reference in the revised manuscript.
Comments:
Else, as a minor comment, references are not set as per journal format.
Response:
We appreciate for the reviewer’s kind reminder. We have standardized the reference format.

Reviewer 2 Report
Liu et al. have investigated preparation of Ag3PO4/g-C3N4@MoS2 photocatalysts and its photocatalytical performance and mechanisms during degradation of 2-amino-4-acetamidoanisole. The work was conducted in depth with variety of characterizations. Controls were used, too. The data were collected and presented carefully. The conclusions are well supported. I have few minor comments:
- Figure 1 describes the morphology of the constituent phases and the composite. The composite is described as Ag3PO4 dispersed on g-C3N4@MoS2, but it is hard to visualize g-C3N4@MoS2. Can the authors use a simple cartoon figure to illustrate the morphology?
- It would have been nicer to label the XRD peaks with Miller indices in Figure 2, although major ones are mentioned in the text.
- The FTIR peak assignments are not supported by references and some are not certain.
- In the optical gap expression, Equation 2, n is not defined. The identification of the gaps as direct or indirect was not stated.
- The XPS data in Figure 5(d) is great and the valence band energy positions are accurately determined. However, what is the low energy shoulder band next to the major band? It does not look like the Boltzmann tail.
- In Figure 9, we see the large band gap of g-C3N4 (3.08 eV) can rate-limit the overall photocatalysis due to low spectral match of absorption with solar excitation. The authors did not comment on this.
Thanks
Author Response
Responses to the Editor and Reviewers’ Comments
Manuscript ID: IJERPH-1579565
Title: “Visible light based Ag3PO4/g-C3N4@MoS2 for highly efficient degradation
of 2-amino-4-acetylaminoanisole (AMA) from printing and dyeing wastewater”
Author(s): Hong Liu*, Houwang Chen, Ning Ding
Reviewer #2:
Comments:
Liu et al. have investigated preparation of Ag3PO4/g-C3N4@MoS2 photocatalysts and its photocatalytical performance and mechanisms during degradation of 2-amino-4-acetamidoanisole. The work was conducted in depth with variety of characterizations. Controls were used, too. The data were collected and presented carefully. The conclusions are well supported. I have few minor comments:
Figure 1 describes the morphology of the constituent phases and the composite. The composite is described as Ag3PO4 dispersed on g-C3N4@MoS2, but it is hard to visualize g-C3N4@MoS2. Can the authors use a simple cartoon figure to illustrate the morphology?
Response:
We appreciate for the reviewer’s comments. As shown in the figure below, g-C3N4 and MoS2 are lamellar materials. The bottom sheet material is g-C3N4, which is covered with MoS2, and finally Ag3PO4 wraps them on top.
Comments:
It would have been nicer to label the XRD peaks with Miller indices in Figure 2, although major ones are mentioned in the text.
Response:
We appreciate for the reviewer’s comments. The XRD peaks have been labeled in Figure 2.
Comments:
The FTIR peak assignments are not supported by references and some are not certain.
Response:
We appreciate for the reviewer’s comments. We have added references to support FTIR peak assignments.
[38]. Zhang, Z.Z.; Pan, ZW.; Guo, Y.F Wong. P.K.; Zhou, X.J.; Bai, R.B. In-situ growth of all-solid Z-scheme heterojunction photocatalyst of Bi7O9I3 /g-C3N4 and high efficient degradation of antibiotic under visible light. Applied Catalysis B: Environmental, 2020, 261.
Comments:
In the optical gap expression, Equation 2, n is not defined. The identification of the gaps as direct or indirect was not stated.
Response:
We appreciate for the reviewer’s comments. According to (αhv)1/n = A(hv– Eg), where α is the absorption index, h is Planck's constant, v is the frequency, Eg is the semiconductor band gap, and A is a constant. Among them, n is related to the type of semiconductor, n of direct bandgap semiconductor is 1/2, and n of indirect bandgap semiconductor is 2. We have added the description in line 310.
Comments:
The XPS data in Figure 5(d) is great and the valence band energy positions are accurately determined. However, what is the low energy shoulder band next to the major band? It does not look like the Boltzmann tail.
Response:
We thank the reviewer for pointing this out, In this experiment, we have not yet figured out the specific reason for the formation of the low energy shoulder band, and we will conduct in-depth research on it in future experiments to obtain a reasonable explanation.
Comments:
In Figure 9, we see the large band gap of g-C3N4 (3.08 eV) can rate-limit the overall photocatalysis due to low spectral match of absorption with solar excitation. The authors did not comment on this.
Response:
We appreciate for the reviewer’s comments. We have added the following in the revised manuscript in line 381.
Although the large band gap of g-C3N4 (3.08 eV) will limit the rate of the whole photocatalysis, its suitable band gap width is just enough to form a Z-type heterojunction composite with Ag3PO4 and MoS2, and meanwhile, its wide band gap is beneficial to electrons transfer with the reduce of recombination of electrons and holes, which can enhance the photocatalytic effect.

Reviewer 3 Report
The presented Manuscript entitled “Visible light based Ag3PO4/g-C3N4@MoS2 for highly efficient degradation of 2-amino-4-acetylaminoanisole (AMA) from printing and dyeing wastewater” relates the photocatalyst synthesis and perfor- 12 mance and mechanism of degradation of 2-amino-4-acetamidoanisole (AMA). In addition, the authors described the catalyst characteriztion by different techniques.
In my opinion the manuscript is well written, presents interesting data. However, some points can be better clarified.
Is the catalyst core shell Ag3PO4/g-C3N4@MoS2? If yes, how can this be said?
Please add the purity of reagents to item 2.1.
What is the band gap of the Ag3PO4/g-C3N4@MoS2 catalyst?
Is the adsorption included in figure 6 ? it is not clear.
How much does the catalyst adsorb? Why 30 min for the adsorption process?
Author Response
Responses to the Editor and Reviewers’ Comments
Manuscript ID: IJERPH-1579565
Title: “Visible light based Ag3PO4/g-C3N4@MoS2 for highly efficient degradation
of 2-amino-4-acetylaminoanisole (AMA) from printing and dyeing wastewater”
Author(s): Hong Liu*, Houwang Chen, Ning Ding
Reviewer #3:
Comments:
The presented Manuscript entitled “Visible light based Ag3PO4/g-C3N4@MoS2 for highly efficient degradation of 2-amino-4-acetylaminoanisole (AMA) from printing and dyeing wastewater” relates the photocatalyst synthesis and performance and mechanism of degradation of 2-amino-4-acetamidoanisole (AMA). In addition, the authors described the catalyst characterization by different techniques. In my opinion the manuscript is well written, presents interesting data. However, some points can be better clarified.
Is the catalyst core shell Ag3PO4/g-C3N4@MoS2? If yes, how can this be said?
Response:
We appreciate for the reviewer’s comments. As shown in the figure below, g-C3N4 and MoS2 are lamellar materials. The bottom sheet material is g-C3N4, which is covered with MoS2, and finally Ag3PO4 wraps them on top. From the SEM image, the composite material should be a sandwich structure.
Comments:
Please add the purity of reagents to item 2.1.
Response:
We appreciate for the reviewer’s comments. Reagent purity has been supplemented in Item 2.1.
Comments:
What is the band gap of the Ag3PO4/g-C3N4@MoS2 catalyst?
Response:
We appreciate for the reviewer’s comments.
(1) The calculation of the band gap of the catalyst is to analyze the possible electron flow direction, energy band structure, etc. on the composite material, so it is not necessary to test the band gap of Ag3PO4/g-C3N4@MoS2.
(2) Before using the XPS data to calculate the valence band, it is necessary to judge whether the material is a P-type material or an N-type material, and then according to (αhv)1/n = A(hv– Eg), where α is the absorption index and h is the Plane g constant, v is the frequency, Eg is the semiconductor band gap, and A is a constant. Among them, n is related to the type of semiconductor, n of direct bandgap semiconductor is 1/2, and n of indirect bandgap semiconductor is 2. It is not clear yet what kind of band gap semiconductor the composite material is, so the measured XPS data could not be brought into the formula for calculation, and the band gap of Ag3PO4/g-C3N4@MoS2 was not measured.
Comments:
Is the adsorption included in figure 6 ? it is not clear.
Response:
We appreciate for the reviewer’s comments. The adsorption is included in Figure 6, and the first 30 minutes indicate the adsorption process of the catalyst, mentioned in line 324-325.
Comments:
How much does the catalyst adsorb? Why 30 min for the adsorption process?
Response:
We appreciate for the reviewer’s comments. The adsorption capacity of the catalyst is small, about 10%. In our previous experiments, it was found that the catalyst could reach the adsorption saturation state in about 10 minutes. Increasing the time to 30 minutes is to observe whether the catalyst will desorb and ensure that the adsorption equilibrium is reached. In the previous literature, we found that most researchers set the adsorption time to 30 minutes to ensure that the catalyst reached the adsorption equilibrium, thereby reducing the effect of adsorption on the photocatalytic experiments in the subsequent experiments.

Round 2
Reviewer 1 Report
The researchers have addressed all of my queries. The manuscript is good enough to be published in present form.